# Characterisation and Outcomes of Patients with Solid Organ Malignancies Admitted to the Intensive Care Unit: Mortality and Impact on Functional Status and Oncological Treatment

**DOI:** 10.3390/diagnostics14070730

**Published:** 2024-03-29

**Authors:** Marta García de Herreros, Juan Carlos Laguna, Joan Padrosa, Tanny Daniela Barreto, Manoli Chicote, Carme Font, Ignacio Grafiá, Lucía Llavata, Elia Seguí, Albert Tuca, Margarita Viladot, Carles Zamora-Martínez, Sara Fernández-Méndez, Adrián Téllez, Josep Maria Nicolás, Aleix Prat, Pedro Castro-Rebollo, Javier Marco-Hernández

**Affiliations:** 1Medical Oncology Department, Hospital Clínic de Barcelona, C/Villarroel 170, 08036 Barcelona, Spain; garciadehe@clinic.cat (M.G.d.H.); laguna@clinic.cat (J.C.L.); padrosa@clinic.cat (J.P.); mchicote@clinic.cat (M.C.); cfont@clinic.cat (C.F.); grafia@clinic.cat (I.G.); llavata@clinic.cat (L.L.); segui@clinic.cat (E.S.); atuca@clinic.cat (A.T.); viladot@clinic.cat (M.V.); czamora@clinic.cat (C.Z.-M.); alprat@clinic.cat (A.P.); 2Translational Genomics and Targeted Therapies in Solid Tumors, Institut d’Investigacions Biomèdiques August Pi i Sunyer (IDIBAPS), 08036 Barcelona, Spain; 3Medical Intensive Care Unit, Internal Medicine Department Hospital Clínic de Barcelona, C/Villarroel 170, 08036 Barcelona, Spain; sfernanm@clinic.cat (S.F.-M.); tellez@clinic.cat (A.T.); nicolas@clinic.cat (J.M.N.); pcastro@clinic.cat (P.C.-R.); 4Radiation Oncology Department, Hospital Clínic de Barcelona, C/Villarroel 170, 08036 Barcelona, Spain; barreto@clinic.cat

**Keywords:** neoplasms, critical care, prognosis, intensive care unit, hospital mortality, hypoalbuminemia, functional status, cachexia

## Abstract

Background: Despite the increasing number of ICU admissions among patients with solid tumours, there is a lack of tools with which to identify patients who may benefit from critical support. We aim to characterize the clinical profile and outcomes of patients with solid malignancies admitted to the ICU. Methods: Retrospective observational study of patients with cancer non-electively admitted to the ICU of the Hospital Clinic of Barcelona (Spain) between January 2019 and December 2019. Data regarding patient and neoplasm characteristics, ICU admission features and outcomes were collected from medical records. Results: 97 ICU admissions of 84 patients were analysed. Lung cancer (22.6%) was the most frequent neoplasm. Most of the patients had metastatic disease (79.5%) and were receiving oncological treatment (75%). The main reason for ICU admission was respiratory failure (38%). Intra-ICU and in-hospital mortality rates were 9.4% and 24%, respectively. Mortality rates at 1, 3 and 6 months were 19.6%, 36.1% and 53.6%. Liver metastasis, gastrointestinal cancer, hypoalbuminemia, elevated basal C-reactive protein, ECOG-PS greater than 2 at ICU admission, admission from ward and an APACHE II score over 14 were related to higher mortality. Functional status was severely affected at discharge, and oncological treatment was definitively discontinued in 40% of the patients. Conclusion: Medium-term mortality and functional deterioration of patients with solid cancers non-electively admitted to the ICU are high. Surrogate markers of cachexia, liver metastasis and poor ECOG-PS at ICU admission are risk factors for mortality.

## 1. Introduction

Admissions of patients with solid organ malignancies to the Intensive Care Unit (ICU) are increasing worldwide. This change is mainly due to the rising incidence of cancer and continuous advances in oncological care [1]. It has been reported that 5% of patients with solid organ cancer will require admission to an ICU during the first two years after diagnosis [2]. Although elective admission for monitoring after programmed oncologic surgery is the most frequent reason for ICU admission in patients with cancer, sepsis and acute respiratory failure are common causes of unplanned ICU admissions [3].

Traditionally, admission of patients with cancer to the ICU has been discouraged due to high mortality rates, high healthcare costs and poor prognosis [4,5]. However, new cancer treatments and early cancer detection have led to longer survival of oncologic patients, including those with advanced disease. These circumstances make decision-making more challenging and can even pose ethical concerns. Such concerns are varied and have been addressed in the literature. Some examples involve unreasonable therapeutic obstinacy [6], limited resources and ICU beds [7], the use of ICU support for a limited time period (ICU trial) [8], the accuracy of triage criteria for the selection of ICU candidates [9], decisions regarding treatment limitation [10] and doubts about the increase in the survival of critically ill patients in recent years [11]. Furthermore, data from several recent studies support an improvement in the survival rates of critically ill cancer patients [12,13,14,15,16,17]. The increase in the complexity of disease and the number of patients with cancer, as well as the rapid changes that have been taking place in recent years in relation to diagnostic and therapeutic strategies in oncology, make close collaboration between experts in oncology and intensive care especially necessary [18].

When patients with cancer become critically ill due to an acute medical condition, physicians must make decisions, usually under the pressure that the passage of time can make the patients’ conditions worse. When considering ICU admission for patients with cancer, clinicians may take into account the severity of the acute medical condition and the prognosis of the neoplasm [19]. Mistakes regarding whether a patient would or would not benefit from being admitted to an ICU can lead to losing options for cure or long survival in some patients with cancer, or, at the opposite end, to applying futile treatments that cause unnecessary suffering at the end of life. Generally, ICU admissions are appropriate when the critical condition is potentially reversible with a reasonable probability, the oncological prognosis involves a life expectancy long enough to justify aggressive therapies, other comorbidities do not advise otherwise, and the patient does not decline intensive-care therapies after discussion with a trained physician, if possible [20]. Multiple risk factors for mortality have already been defined in critically ill patients with cancer. These include factors related to the patient (age [16,20] and Eastern Cooperative Oncology Group Performance Status [ECOG-PS] scale prior to hospitalisation [2,20,21]), factors related to the cancer (type of malignancy [22], tumour stage [16,20,22], neutropenia at admission [16] and chemotherapy administered in the previous month [16]) and factors related to the acute cause of deterioration (late ICU admission [16], reason for ICU admission [2,22], number of organ failures [16,23,24], invasive mechanical ventilation [2,22], presence of invasive fungal infection [20] and septic shock [22,25]).

However, these studies have some limitations. Some of them are non-homogeneous retrospective studies that include ICU admissions after elective procedures and mixing patients with haematological malignancies and those with solid organ cancers [26]. Furthermore, they tend to focus on factors that influence the short-term mortality, but there are limited data on medium-term mortality and, especially, on the impact of ICU admission on functional status and oncological treatment [27,28].

A systematic review analysed the results of published articles from 2000 that studied the survival of patients with solid organ cancer who required admission to the ICU and revealed a marked methodological heterogeneity in the variables analysed and the characteristics of the patients, as well as a very wide range of mortality [2]. One of the best articles published about the outcomes of patients with cancer, particularly those with solid organ neoplasms who were admitted In the ICU for an unplanned reason, analysed a 10-year period in a French hospital and had objectives similar to those proposed in our study. As a novel aspect, it addressed what how cancer treatments progressed after the acute complications that had led the patients to the ICU has been resolved [27].

In our study, we aimed to analyse the characteristics and outcomes of patients with solid organ malignancies non-electively admitted to the ICU in a leading Spanish hospital. The primary objective was to assess in-hospital mortality and 3- and 6-month mortality. Prolonging survival is not the only important issue for patients with cancer; it is also important that this time be of the highest quality possible. Furthermore, this question represents an important knowledge gap in the literature. That is why the evaluation of changes in functional status was also proposed as an objective. In fact, it may constitute a surrogate marker of quality of life. Thus, the secondary objectives were the impact of ICU admission on functional status and oncological treatment of ICU survivors (chemotherapy, immunotherapy and targeted therapies, among others), as well as identification of prognostic factors associated with ICU outcomes in this population.

## 2. Material and Methods

### 2.1. Study Design

This is a retrospective observational study that included all patients with solid organ cancers non-electively admitted to the ICU of the Hospital Clínic of Barcelona between January and December 2019. The period was selected to avoid any bias introduced by the SARS-CoV-2 pandemic. The study protocol was approved by the local Ethics Committee (approval code HCB/2020/1377).

### 2.2. Patients: Inclusion and Exclusion Criteria

All patients with solid organ cancers aged 18 years or older non-electively admitted to the ICU during 2019 were included in the study. This category includes patients undergoing neoadjuvant, adjuvant or metastatic cancer treatment, as well as those receiving best supportive care and patients who newly diagnosed at the time of ICU admission. Patients admitted during the immediate post-tumour-surgery period or for monitoring after planned procedures and patients who had been cancer-free for at least 5 years since diagnosis were excluded.

### 2.3. Data Collection

The data was collected and anonymized by the research team, as authorized by the Ethics Committee, from electronic hospital medical records. Demographic data, comorbidities and functional condition (Barthel [29], Karnofsky [30] and ECOG-PS [31] scores), along with cancer-related information, were recorded. Laboratory tests such as neutropenia, basal C-reactive protein (CRP) and albumin were recorded. Hypoalbuminemia was defined as albumin <32 g/L, and elevated basal CRP was defined as >0.5mg/dL in an outpatient blood test obtained the month before ICU admission. Data regarding ICU stay were also collected, such as the reason for admission, the department of origin, the severity of the acute condition (Acute Physiology and Chronic Health disease Classification System II [APACHE-II] [32] and Sequential Organ Failure Assessment [SOFA] [33] scores) and the type of support therapy received (ventilatory support, vasoactive drugs and kidney replacement therapy, among others). APACHE II score results range from 0 to 67 points, and SOFA score results range from 0 to 24 points, with higher values reflecting a more severe condition and higher mortality risk. We also documented changes in therapy goals and end-of-life decisions. We assessed data regarding the patients’ conditions at discharge and clinical outcomes, such as modifications to or discontinuations of cancer treatment and the need for new hospital and ICU admissions, as well as 1-, 3-, and 6-month mortality. Each ICU admission was considered regardless of whether it was a new admission or a readmission of a patient who had previously been admitted during 2019.

### 2.4. Statistical Analysis

The R language, version 4.2.1, was used for statistical analyses. Categorical variables were reported as percentages. Comparisons between groups were performed using the chi square test or Fisher’s exact test for dichotomous variables. Continuous variables were reported as the mean (standard deviation [SD]) or median (interquartile range [IQR]). Comparisons were made using Student’s *t*-test, the Mann–Whitney *U* test or analysis of variance (ANOVA), as appropriate. Statistical significance was considered at *p* < 0.05. Survival was evaluated using the Kaplan-Meier method, and variables were adjusted using logistic regression models.

## 3. Results

### 3.1. Baseline Characteristics of the Patients

A total of 97 ICU admissions of 84 patients with solid malignancies were included, with a median follow-up of 14.3 months. Forty-three patients (51.2%) were men, and the mean age was 62 years old. The most common cancers among the 84, in descending order, included lung cancer (19 patients; 22.6%), colorectal cancer (14 patients; 16.7%), breast cancer (11 patients; 13%) and renal cell carcinoma (6 patients; 7.1%). Fewer than five patients suffered from the remaining neoplasms represented in the cohort (head and neck, gynaecological, pancreatic, gastric, central nervous system and prostate cancers, as well as malignant melanoma and cholangiocarcinoma). The baseline characteristics of the patients admitted to the ICU are shown in Table 1. The functional status of the patients and the status of the oncological disease was reevaluated for each ICU admission in those patients who were admitted more than once during 2019 due to the variability in these factors throughout time.

At the time of each ICU admission, most patients suffered from stage IV neoplasms (77 cases; 79.4%). Sixteen patients suffered from stage III neoplasms (16.5%), while only four had stage I or II neoplasms (4.1%). Seventy-three patients (75.3%) were receiving oncological treatment before ICU admission, mainly chemotherapy (49 patients; 50.5%), followed in decreasing order by targeted therapies (26 patients; 26.8%), immunotherapy (12 patients; 12.4%) and hormone therapy (4 patients; 4.1%). A total of 82.6% of these latter patients had received oncological treatment in the 30 days prior to ICU admission.

### 3.2. Characteristics of ICU Admission and Mortality

The main causes of admission to the ICU were respiratory failure (37 admissions; 38%) and septic shock (32 admissions; 34%). The remaining 28% of ICU admissions were indicated for monitoring purposes related to other conditions, such as severe bleeding, dyselectrolythemia, neurological deterioration and high-risk pulmonary embolism. The mean APACHE II and SOFA scores at admission were 14.5 and 4.7, respectively. Further data regarding ICU admission are shown in Table 2.

Cancer was diagnosed during the same hospital admission in 12.4% of the patients (half of them in the ICU). Five patients received chemotherapy during their ICU stays.

The median ICU length of stay (LoS) was 4 days (IQR 3), and the in-hospital LoS was 22 (IQR 9) days, longer than the in-hospital LoS for non-critical patients in our centre (10.5 days). In 31% of the cases, changes in therapy goals and discussions of end-of-life options occurred during hospitalisation.

The median overall survival of the entire cohort was 168 days (95% confidence interval [CI] 127.232). Sixty-three out of the eighty-four patients included in the study were dead 12 months after admission to the ICU, which represents a 75% 1-year mortality. In-ICU and in-hospital mortality were 9.4% (8 patients) and 24% (20 patients), respectively. Mortality rates after ICU admission were 15.5% (13 patients) at 1 month, 35.7% at 3 months (30 patients) and 56% at 6 months (47 patients). Further data are given in Appendix A
Figure A1.

### 3.3. Variables Associated with Mortality

Univariate analysis showed that the presence of liver metastasis or gastrointestinal cancers was associated with a higher in-hospital mortality. Interestingly, elevated basal CRP and hypoalbuminemia were also associated with a worse prognosis and higher in-hospital and 3-month mortality. Additionally, patients with an ECOG-PS greater than 2 at the moment of ICU admission, those admitted to the ICU from a ward compared to those admitted from the emergency department and patients diagnosed with cancer during the same hospital admission had higher rates of in-hospital mortality. Regarding the severity of the acute illness, patients with an APACHE II score greater than 14 points had a higher in-hospital mortality, while no significant increased risk was found in patients with a SOFA score greater than 4 points. There were no differences in mortality in patients who required vasoactive support or mechanical ventilation. Table 3 shows the risk factors evaluated for in-hospital mortality.

Unadjusted Kaplan-Meier survival curves for relevant variables associated with increased mortality are shown in Figure 1.

After adjusting for age, gender, reason for ICU admission and the APACHE II score, three variables were found to predict in-hospital mortality. These variables were an ECOG-PS at ICU admission greater than 2, basal CRP levels greater than 0.5 mg/dL and admission to the ICU from a ward rather than the emergency room, with odds ratios (OR) of 6.96 (95% CI 2.22–21.75), 14.96 (95% CI 1.82–122.89) and 6.17 (95% CI 1.42–29.85), respectively. They were also associated with a higher 3-month mortality, with an OR for an ECOG-PS greater than 2 of 5.01 (95% CI 1.86–13.44), for CRP of 6.04 (95% CI 1.89–19.35) and being admitted from a ward the OR was 4.54 (95% CI 1.41–14.6). Patients diagnosed with cancer during ICU admission had a 50% in-hospital mortality rate.

Unlike patients with stage IV solid organ malignancies, patients with non-metastatic disease and those receiving treatments with curative intent had the lowest mortality rates (77% alive at 2 years).

### 3.4. Impact of ICU Admission on Functional Status and Oncological Treatment

At hospital discharge, functional status was severely affected. Seventy-eight (79%) patients had an ECOG-PS greater than 1 at hospital discharge, compared to only 25.5% of patients one month before ICU admission. Figure 2 illustrates this ECOG-PS deterioration (*p* < 0.001). Forty percent of the patients had a Karnofsky index less than 40% at discharge, a number significantly different compared to the score at admission (*p* = 0.01). The median Barthel Index also decreased from 100 the month before admission to the ICU to 70 upon discharge (40% of patients with a score less than or equal to 60), and this change was also statistically significant (*p* < 0.001).

Oncological treatment was definitively discontinued in 40% of the patients, whereas dose reduction or delay in treatment was necessary in 56.7% of patients. In 18.6% of the cases, treatment was changed due to cancer progression or clinical deterioration. A total of 40 patients (41%) were readmitted to the hospital within 180 days after their initial ICU admission, and eleven (11.3%) required admission to a community health centre or hospice upon hospital discharge. Further data are shown in Table 4.

## 4. Discussion

In the present study, we found that patients with solid organ cancer who were non-electively admitted to the ICU have high short and medium-term mortality, as well as a significant deterioration in functional status, a high rate of hospital readmission and a frequent need for changes in oncological treatment. Studies specifically designed to examine the outcomes of patients with solid organ cancers in the ICU are scarce and mostly single-centre, non-homogeneous and retrospective. A prognostic score that considers the type and stage of the cancer and organ support received in the ICU, with total score from 1 to 12, was designed with the objective of predicting 120-day post-ICU mortality, but it has not yet been validated [22].

The results of our study are consistent with those published in recent years. Some of the mortality risk factors found in our study have been previously described by other authors [2,16,21,26,27,28]. According to previous data, we have confirmed that cancer stage, gastrointestinal malignancies and the APACHE II score at ICU admission correlate with higher mortality rates. Furthermore, we also identified that the presence of liver metastasis, surrogate markers of cachexia (hypoalbuminemia and elevated basal CRP levels) and bedside ECOG-PS assessment upon ICU admission were also predictors of mortality. To the best of our knowledge, these prognostic factors have not been widely reported, and they can be easily evaluated during clinical practice.

Cancer cachexia is a multifaceted condition characterized by malnutrition, loss of muscle mass and anorexia and is often accompanied by systemic inflammation in individuals with advanced cancer [34]. Although it has been associated with unfavourable outcomes [35], its role as a risk factor for mortality has not been investigated in patients admitted to the ICU. Certain surrogate markers of cachexia [36], such as hypoalbuminemia and elevated basal CRP levels during outpatient follow-up, have been correlated with increased mortality.

In our cohort, an ECOG-PS score greater than 2 at ICU admission was associated with higher in-hospital and 3-month mortality after adjusting for age, gender, APACHE-II score and reason for admission. Although it may seem obvious, this finding emphasizes the relevance of bedside clinical assessment when evaluating a critically ill patient with cancer. A previous study that was not focused on cancer alone reported that a reduced ECOG-PS score was associated with worse outcomes, independent of other chronic health markers in a global cohort of critically ill patients [21].

An additional interesting condition associated with worse outcomes in our cohort was the timepoint of ICU admission. Patients who developed an acute critical deterioration during hospitalization and required ICU admission showed higher mortality rates compared to those admitted directly from the emergency department. This finding aligns with recent meta-analytical data [37], where patient origin (hospitalisation ward vs. emergency department) impacted mortality rates. The suggested rationale behind this observation is that early ICU admission might directly influence mortality outcomes and that management of critically ill patients is faster in the emergency department than in a hospitalisation ward. Furthermore, patients hospitalized for a prolonged period often experience deterioration in general condition, cachexia and loss in muscle mass, all of which could affect prognosis.

These previously discussed risk factors could help clinicians identify patients who are likely to present higher in-hospital mortality rates. The appropriateness of therapeutic goals and the minimization of invasive procedures in patients with higher risk should be considered in weighing the pros and cons of ICU admission. However, some patients with advanced cancer, such as those showing complete or sustained partial response to ongoing cancer treatment, may also benefit from ICU admission. These patients were underrepresented in our cohort and deserve further study. On the other hand, patients with cancers in localized stages, patients who are receiving treatment with curative intention, and patients whose tumours exhibited complete responses to treatment mostly presented favourable outcomes and long-term survival rates. Furthermore, these patients did not have any surrogate markers of cachexia, and their ECOG-PS scores before ICU admission were less than or equal to 2. This unique subgroup of patients should not pose any dilemma regarding indications and potential benefits of admission to the ICU.

Mortality is typically the primary outcome of most studies evaluating ICU outcomes. However, many authors have pointed out the need to look for beyond the ICU admission and survival, with the goal of achieving adequate long-term multidimensional physical and mental health for surviving patients and their families [38]. Quality of Life (QoL) has always been difficult to evaluate in retrospective studies because it is not an easily measurable parameter. Studies focusing on the QoL of patients with cancer after ICU admission [26,28,39] have found that it decreases significantly 3 months after ICU discharge, especially in older patients, patients with a previous poor ECOG-PS score, and patients with severe comorbidities. Relatedly, in our study, ICU admission was found to be associated with factors that may be related to reductions in QoL, such as hospital readmission; discontinuation of cancer treatment; significant deterioration in the ECOG-PS score and Barthel and Karnofsky Indices; and the need for admission to a community health centre or hospice at the time of discharge.

The present study has several limitations. Being retrospective, it can only generate hypotheses, and the findings must be confirmed and validated in prospective studies. The finding of associations and the identification of potential risk factors is valuable, but causality cannot be established. The main limitation of the study, which must be highlighted, is the limited sample size and the limited representation of less common solid organ cancers. Notably, most patients had stage IV disease, and more than half had received chemotherapy within the previous 30 days. These two facts might have led to worse outcomes and skewed the results. Biases related to data collection and loss of information in clinical records could have occurred.

However, the strengths of this study lie in its analysis of the effects of ICU admission on outcomes beyond mortality, such as functional status, QoL, readmission rates and changes in oncological treatment. These findings may be useful in deciding treatment goals for each patient at the time of ICU admission.

## 5. Conclusions

It is a real challenge for physicians to identify which patients with cancer would benefit from ICU admission. In our cohort, a quarter of the patients with cancer admitted to the ICU died during the same hospital admission and more than a third died within 3 months of ICU admission. Surrogate markers of cachexia, liver metastasis and a poor ECOG-PS score at the time of ICU admission were found to be newly described risk factors for increased mortality. Furthermore, ICU admission had a significant impact on functional status and subsequent oncological treatment among survivors.

Objective, multivariate, and integrative tools for adequate selection of ICU candidates among patients with cancer are needed. There is an urgent need for large prospective studies assessing medium- and long-term outcomes after ICU admission and, not least, assessing the quality of life of surviving patients.

## Figures and Tables

**Figure 1 diagnostics-14-00730-f001:**
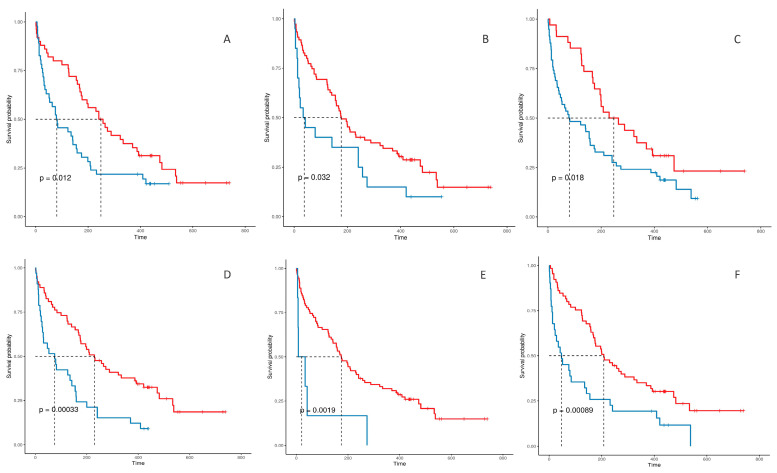
Kaplan Meier survival curves by variable. (**A**) compares admitted from emergency department (ER) with those admitted from hospitalisation wards; (**B**) compares patients with albumin <32 g/L; (**C**) compares patients with basal C-reactive protein (CRP) >0.5 mg/dL with those with basal CRP ≤0.5 mg/dL; (**D**) reflects lower survival in patients with liver metastasis; (**E**) compares patients diagnosed during the ICU stay with those with previous diagnosis of cancer; (**F**) compares patients with ECOG-PS score at admission >2 with those with ECOG-PS score ≤2.

**Figure 2 diagnostics-14-00730-f002:**
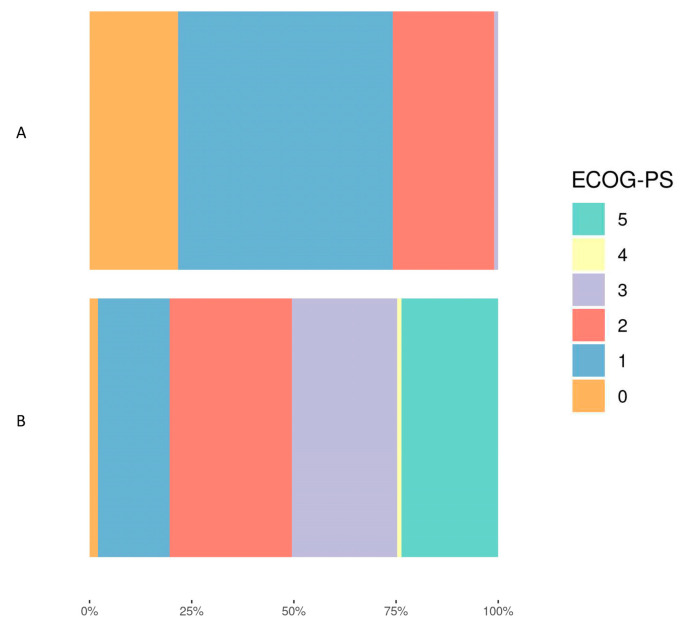
The figure represents the ECOG-PS deterioration during admission. (**A**) shows the distribution of ECOG-PS of the patients before ICU admission and (**B**) reflects the ECOG-PS of the patients at discharge.

**Table 1 diagnostics-14-00730-t001:** Patients’ baseline characteristics before ICU admission.

**Patients**		84
**ICU admissions**		97
**Baseline characteristics (*N* = 84 patients)**		
Age (y), mean (SD)		62 (4.6)
Men, *n* (%)	43 (51.2%)	
Body mass index, mean (SD)		25.6 (5.8)
Significant comorbidities excluding cancer *, *n* (%)	0	23 (27.4%)
	1	29 (34.5%)
	2	12 (14.3%)
	3	12 (14.3%)
	≥4	8 (9.5%)
Polypharmacy ** (%)	38 (45.2%)	
Presence of at least 1 geriatric syndrome ***, *n* (%)	42 (50%)	
**Functional status evaluated before each ICU admission (*N* = 97 ICU admissions)**		
Barthel Index 1 month before admission, median (range)	100 (100–20)	
ECOG-PS **** 1 month before ICU admission, *n* (%)	0	21 (21.6%)
	1	51 (52.6%)
	≥2	25 (25.8%)
Karnofsky Index 1 month before admission, median (range)	80 (100–20)	
**Status of oncological disease evaluated before each ICU admission (*N* = 97 admissions)**		
Stage IV cancer, *n* (%)	77 (79.4%)	
Ongoing anti-tumour treatment, *n* (%)	73 (75.3%)	
Curative intention of treatment, *n* (%)	16 (16.5%)	

* Main comorbidities include hypertension diabetes, and chronic organ disfunctions; ** Polypharmacy is defined as 5 or more drugs used as chronic medications, excluding cancer treatment medicines; *** Geriatric syndromes: immobility, pressure ulcers, falls, need for orthoses for movement, urinary or faecal incontinence, cognitive impairment, polypharmacy (defined as 5 or more chronic drugs on admission), hearing or visual loss, depression, constipation or faecal impactions and confusional syndrome during hospitalization; **** ECOG-PS; performance status according to the Eastern Cooperative Oncology Group.

**Table 2 diagnostics-14-00730-t002:** ICU admission characteristics and ICU requirements (*N* = 97).

**Main Cause of ICU Admission**	***N* (%)**
Respiratory failure	37 (38%)
Sepsis or septic shock	32 (34%)
Cardiogenic instability or pulmonary embolism	8 (8.2%)
Severe bleeding	8 (8.2%)
Acute kidney injury or dyselectrolythemia	5 (5.2%)
Post-operative (non-elective surgeries)	5 (5.2%)
**ICU scales at admission**	**Mean (SD)**
SOFA *	4.7 (2.8)
APACHE II **	14.5 (5.9)
**ICU requirement during admission**	***N* (%)**
Need for any type of respiratory support	72 (74.2%)
Type of respiratory support:	
Venturi or nasal cannula	37 (38.1%)
High-flow nasal cannula	17 (17.5%)
Non-invasive mechanical ventilation	4 (4.1%)
Invasive mechanical ventilation	14 (14.4%)
Need for vasoactive drugs	37 (38.1%)
Need for surgical procedure	13 (13.4%)
Need for kidney-replacement therapy	6 (6.2%)

* SOFA, Sequential Organ Failure Assessment score; ** APACHE II, Acute Physiology and Chronic Health Evaluation II score.

**Table 3 diagnostics-14-00730-t003:** Risk factors for in-hospital mortality.

Variable	Frequency (%)	Odds Ratio (CI95%) *
Female	48.8%	0.41 (0.15–1.12)
Cancer diagnosis in the same hospitalisation	12.4%	**4** (1.15–13.96)
Cancer diagnosis in ICU	6.2%	3.55 (0.66–18.96)
Metastatic disease (stage IV)	79.5%	1.99 (0.51–7.29)
Treatment received <30 days	42.9%	1.32 (0.26–6.76)
Presence of liver metastasis	35%	**3.28** (1.25–8.63)
Hypoalbuminemia (<32 g/L)	21.6%	**4.34** (1.53–12.32)
Gastric cancer	4.1%	All patients died during admission.
ECOG-PS >2 at the moment of ICU admission	33%	**8.29** (2.91–23.60)
Admission from ward	48.5%	**4.16** (1.47–11.76)
APACHE II score at ICU admission >14	42.3%	**3.46** (1.30–9.24)
SOFA score at ICU admission >4	42.3%	2.14 (0.83–5.52)
Need for vasoactive support	38%	1.13 (0.44–2.91)

* CI, confidence interval. Values in bold were statistically significant.

**Table 4 diagnostics-14-00730-t004:** Outcomes beyond mortality: length of hospital stay, readmission rates, changes in functional status and treatment modifications (*N* = 97 ICU admissions).

Outcomes beyond Mortality	Patients (%)
ICU length of stay in days (median, IQR)	4 (2–6)
Length of hospital stay in days (median, IQR)	25 (11–34)
Changes in therapeutic goals (*n*, %)	30 (31%)
Need for admission to a community health centre or hospice at discharge (*n*, %)	11 (11.3%)
**Readmission rates**	
ICU readmission at 6 months (*n*, %)	11 (11.3%)
Hospital readmission at 6 months (*n*, %)	40 (41%)
**Changes in functional status**	
Barthel index (median, IQR)	
ICU admission	90 (IQR 80–94)
Discharge	70 (IQR 62–80)
Karnofsky index (median, IQR)	
ICU admission	60 (IQR 45–84)
Discharge	50 (IQR 40–65)
ECOG PS ≥ 2 (*n*, %)	
ICU admission	70 (72%)
Discharge	78 (79%)
**Treatment modifications**	
Definitive discontinuation (*n*, %)	39 (40%)
Changes in line of treatment (*n*, %)	18 (18.6%)
Treatment dose modification (*n*, %)	55 (56.7%)

## Data Availability

Data are contained within the article.

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
