# Peer review of "Characterisation and Outcomes of Patients with Solid Organ Malignancies Admitted to the Intensive Care Unit: Mortality and Impact on Functional Status and Oncological Treatment"

_diagnostics, 2024, doi:10.3390/diagnostics14070730_

Round 1
Reviewer 1 Report
Comments and Suggestions for Authors
Abstract:
Delete (2), (3), (4). They are not the Journal requirement.
What are thoracic malignancies, oncologic treatment, ECOG-PS, APACHE II?
What type of functional status? Physical?
Introduction
What is Eastern Cooperative Oncology 55 Group Performance Status [ECOG-PS] scale?
It is not clear about the knowledge gaps for the study described in this part. Why did the study focus on functional status? What aspect?
Oncological treatment: what types?
Prognostic factors: why did you investigate them? what are they and what did you collect to understand these factors?
Data collection:
Who did data retrieval from the electronic medical records?
Elaborate more about the instruments. How were they interpreted? What did high and low scores indicate?
Results:
A total of 97 samples was too small unless sample size estimation was done.
What were the latter patients?
Readmission rates: due to the same problem?
Discussion:
Rewrite this: “Cancer cachexia is as a complex condition which include malnutrition, loss of muscle mass and anorexia in the presence of systemic inflammation in patients with advanced cancer. It has been associated with poor outcomes but has not been studied as a mortality risk factor in those patients admitted to the ICU. Some surrogate markers for cachexia have been associated with mortality, such as hypoalbuminemia and high basal CRP levels during ambulatory follow-up”
Unclear: ‘Patients admitted from a hospital ward (on the development of a critical condition during hospitalisation) had higher mortality compared 240 to those directly admitted from the emergency department. This agrees with data reported in a recent meta-analysis by Hourmant et al31. The explanation proposed for this finding is that early admission to the ICU when required and appropriate might have a direct impact on mortality outcomes, although the poorer clinical condition of patients who have 244 been admitted for an extended period may also contribute.”. please rewrite.
Quality of life was not your focus. Why did you mention?
Reference list: why there are empty references with numbers?
Others:
Check the format of referencing, including in-text citations and reference list. Please follow the author’s guideline.
Comments on the Quality of English Language
Minor
Author Response
Thank you very much for taking the time to review this manuscript. We have tried to address both the major and minor concerns you pointed. We hope that our proposals resolve them satisfactorily and, therefore, improve the work. Please find the detailed responses below and the corresponding revisions/corrections highlighted changes in the re-submitted files, as well as a clean version of the revised manuscript.
Point-by-point response to Comments:
Comment 1: Abstract: Delete (2), (3), (4). They are not the Journal requirement.
Response 1: Done.
Comment 2: Abstract: What are thoracic malignancies, oncologic treatment, ECOG-PS, APACHE II?
Response 2: We have changed the expression “thoracic malignancies” to “lung cancer”, which is more understandable. Oncological treatment refers to chemotherapy, immunotherapy, targeted therapies for cancer, etc. It is reported in detail in the results section. The ECOG-PS is the acronym for “Eastern Cooperative Oncology Group – Performance Status” scale, a score broadly used in oncology to report general well-being of patients with cancer and determine if they can receive oncological treatment. It is referenced in the introduction and material and methods sections. APACHE II is the acronym for “Acute Physiology and Chronic Health disease Classification System II”, which is a scoring system used to evaluate disease severity commonly used in intensive care units. It is referenced in the material and methods section.
Comment 3: Abstract: What type of functional status? Physical?
Response 3: Thank you for raising this important question. The functional status or performance status of a patient is defined as the level of activity that an individual can perform and their capacity for self-care. The most common scales used in oncology for this purpose are ECOG-PS scale, Karnofksy Index and Barthel Index, which have been referenced in the material and methods section.
Comment 4: Introduction: What is Eastern Cooperative Oncology 55 Group Performance Status [ECOG-PS] scale?
Response 4: The ECOG-PS is the acronym for “Eastern Cooperative Oncology Group – Performance Status” scale, a score broadly used in oncology to report general well-being of patients with cancer and determine if they can receive oncological treatment.
Comment 5: Introduction: It is not clear about the knowledge gaps for the study described in this part. Why did the study focus on functional status? What aspect?
Response 5: We are glad you brought up this question, since it is just one of the features that makes our study different from others. We focus on functional status beyond survival because we believe that for patients with cancer is not only important to live more time, but that this time to be of the highest quality possible. The functional status has been evaluated through the usual scales used in oncology such as ECOG-PS, Karnofksy and the Barthel Index. Moreover, the functional status condition changes and other endpoints can constitute surrogates of quality of life. We have tried to clarify this in the introduction to remark this knowledge gap.
Comment 6: Introduction: Oncological treatment: what types?
Response 6: Oncological treatments are very diverse, including chemotherapy, immunotherapy and targeted therapies (antiangiogenics, tyrosine kinase inhibitors, antibody-drug conjugates, etc.) among others. We have clarified that in the introduction. It is also reflected in the results section.
Comment 7: Introduction: Prognostic factors: why did you investigate them? what are they and what did you collect to understand these factors?
Response 7: In response to this relevant question, multiple prognostic factors for ICU mortality in critically ill patients with cancer have already been defined by other authors. Thus, we wanted to investigate them in our cohort of patients. To understand these factors, we collected variables such as age, ECOG-PS scale prior to hospitalisation, type of malignancy, tumour stage, neutropenia at admission, chemotherapy administered in the previous month, late ICU admission, reason for ICU admission, number of organ failures, invasive mechanical ventilation, among others. Those are reflected in the introduction and material and methods sections.
Comment 8: Data collection: Who did data retrieval from the electronic medical records?
Response 8: Given that it is a retrospective patient cohort, clinical data was collected by the research team itself from electronic medical records. It has been clarified in the material and methods section.
Comment 9: Data collection: Elaborate more about the instruments. How were they interpreted? What did high and low scores indicate?
Response 9: APACHE II and SOFA scores are used in daily routine in intensive care units to quantify the severity of illness in critically ill patients. A high value of either score indicates greater physiological derangement or organ dysfunction, which is associated with increased mortality risk and poorer clinical outcomes. These scores help healthcare providers assess the severity of illness, allocate resources appropriately, and guide clinical decision-making in the ICU setting. In order to clarify that for the interpretation of a high or low result for readers we have expanded information about it in the text. Additionally, all the evaluation scales used have been referenced.
Comment 10: Results: A total of 97 samples was too small unless sample size estimation was done. What were the latter patients?
Response 10: We understand your perspective and agree about the limited sample size. We have emphasized that in the discussion section. Some patients were excluded due to insufficient information in their medical records and all the planned ICU admissions as well (e.g. oncologic surgery). Patients from 2020 onwards were not included to ensure COVID-19 pandemic biases. Despite the modest sample size, we have been able to identify statistically significant mortality predictive factors and changes on functional status. We really believe that the study conclusions are clinically relevant and provide novel information in the scenario of critically ill patients with cancer.
Comment 11: Results: Readmission rates: due to the same problem?
Response 11: Readmission rates were considered for the same problem or for a different one (for example, a different infection, thrombosis, etc.) in the following months.
Comment 12: Discussion: Rewrite this: “Cancer cachexia is as a complex condition which include malnutrition, loss of muscle mass and anorexia in the presence of systemic inflammation in patients with advanced cancer. It has been associated with poor outcomes but has not been studied as a mortality risk factor in those patients admitted to the ICU. Some surrogate markers for cachexia have been associated with mortality, such as hypoalbuminemia and high basal CRP levels during ambulatory follow-up”
Response 12: As suggested, we have clarified this paragraph; it has been rewritten as follows: “Cancer cachexia is a multifaceted condition characterized by malnutrition, muscle mass loss, and anorexia, often accompanied by systemic inflammation in individuals with advanced cancer. While it has been linked to unfavourable outcomes, its role as a mortality risk factor has not been investigated in ICU-admitted patients. Certain surrogate markers for cachexia, such as hypoalbuminemia and elevated basal CRP levels during outpatient monitoring, have been correlated with higher mortality.”
Comment 13: Discussion: Unclear: ‘Patients admitted from a hospital ward (on the development of a critical condition during hospitalisation) had higher mortality compared to those directly admitted from the emergency department. This agrees with data reported in a recent meta-analysis by Hourmant et al. The explanation proposed for this finding is that early admission to the ICU when required and appropriate might have a direct impact on mortality outcomes, although the poorer clinical condition of patients who have 244 been admitted for an extended period may also contribute.”. please rewrite.
Response 13: Thank you for the suggestion; we have tried to clarify this paragraph as well. It has been rewritten as follows: “Patients who developed an acute critical deterioration during hospitalization and needed for ICU admission exhibited higher mortality rates compared to those admitted directly from the emergency department. This finding aligns with recent meta-analytical data, were patient’s origin (hospitalisation ward vs emergency department) impacted on mortality rates. The suggested rationale behind this observation is that early ICU admission might directly influence mortality outcomes and the management of critically ill patients is faster in the emergency department than in a hospitalisation ward. Furthermore, patients hospitalized for an extended period typically experience deterioration of general status, cachexia and loss in muscle mass, all of which could impact on prognosis.”
Comment 14: Discussion: Quality of life was not your focus. Why did you mention?
Response 14: We are glad you brought up this point for further discussion. As mentioned previously, beyond mortality rates, one of our objectives was to assess the deterioration in the quality of life of our patients evaluated by the decline in Barthel, ECOG-PS and Karnofsky scales, the need for community health centre or hospice upon discharge and the readmission rate. We have tried to make that clearer in the introduction and discussion sections.
Comment 15: Reference list: why there are empty references with numbers?
Response 15: That was a problem of editing when the manuscript was adapted to the journal template. It has been solved.
Comment 16: Check the format of referencing, including in-text citations and reference list. Please follow the author’s guideline.
Response 16: Thank you for pointing that out. We have checked them and modified the in-text citations to try to maintain the same criteria for references.
Reviewer 2 Report
Comments and Suggestions for Authors
This is a well conducted study assessing the outcomes and factors associated with various outcomes in solid organ cancer patients admitted in the ICU in a large center. The study is well performed, the manuscript is well written, results are complete and conclusions correspond well to findings. The narrative parts of the study are adequately written and contain sufficient comparisons and evidence to support findings. Limitations and strengths are acknowledged. Some minor comments:
Line 39: I would assume additional reasons for this, therefore I would rephrase as “This is mainly due..”
Line 78: please add Ethics committee approval number if available.
Please define “geriatric syndrome”.
Please align columns and rows in Table 2, as the results aren’t clear.
Last paragraphs of the discussion are quite fragmented, please revise.
Please correct section 5 to “Conclusions”.
Author Response
Thank you very much for taking the time to review this manuscript. We have been very flattered by your comments about the work. Please find the detailed responses below and the corresponding revisions/corrections highlighted in the re-submitted files, as well as a clean version of the revised manuscript.
Point-by-point response to Comments:
Comment 1: I would assume additional reasons for this, therefore I would rephrase as “This is mainly due..”
Response 1: Thank you for pointing this out. We agree with this comment, so we have rephrased the sentence.
Comment 2: please add Ethics committee approval number if available.
Response 2: We have added the Ethics committee approval number to the text.
Comment 3: Please define “geriatric syndrome”.
Response 3: The definition of geriatric syndromes has been added.
Comment 4: Please align columns and rows in Table 2, as the results aren’t clear.
Response 4: Thank you for noticing. It was a mistake when editing the manuscript in the journal template.
Comment 5: Last paragraphs of the discussion are quite fragmented, please revise.
Response 5: The last paragraphs of the discussion have been rephrased to solve fragmentation.
Comment 6: Please correct section 5 to “Conclusions”.
Response 6: Thank you for pointing this out. We have corrected the title of the section 5.
Reviewer 3 Report
Comments and Suggestions for Authors
This article aims to characterize the clinical profile and outcomes of patients with solid malignancies admitted to the ICU. The research itself seems sound, but the quality of the writing could be better:
- The Introduction and Discussion sections are too short. Why are there two Discussion sections (4 and 5)? Provide more background material, and put your research in a wider context.
- There are only 32 references of which 21 are used in three sentences (line 49, 51 and 60). In line 49, perhaps you can list the ethical concerns and use one reference for each. The same for the references in lines 51 and 60. Provide references for ECOG-PS, Barthel and Karnofsky scores. The same for Apache-II and SOFA. Discuss (and reference) studies that are similar to this study.
- Artificial intelligence is having a huge impact on the ICU already, and could help solve several problems that are raised in this paper. This should be discussed in the discussion section.
Minor:
- Please correct the reference numbering.
- Different font in lines 227-230.
- Supplementary figure: either put it in the paper as figure 3, or put it in the supplementary material.
Author Response
Thank you very much for taking the time to review this manuscript We are grateful you found or research sound. We have tried to address both the major and minor concerns you pointed. We hope that our proposals resolve them satisfactorily and, therefore, improve the work. Please find the detailed responses below and the corresponding revisions/corrections highlighted changes in the re-submitted files, as well as a clean version of the revised manuscript.
Point-by-point response to Comments:
Comment 1: The Introduction and Discussion sections are too short. Why are there two Discussion sections (4 and 5)? Provide more background material, and put your research in a wider context.
Response 1: We have extended both sections and provided further background material. As you suggested, that puts the research in a wider context. Section 5 was not a second Discussion section; it was a typing mistake which has been corrected as well.
Comment 2: There are only 32 references of which 21 are used in three sentences (line 49, 51 and 60). In line 49, perhaps you can list the ethical concerns and use one reference for each. The same for the references in lines 51 and 60.
Response 2: We have found this remark very accurate and useful to improve the manuscript, so we have carried out several actions in this regard. First, we have extended the total number of references to better contextualize the work. Second, we have followed your suggestion for the lines addressing the ethical concerns and the same for the references regarding risk factors for mortality. However, we have decided to maintain the references about the improvement in survival rates in the same way since the six articles supported this very same idea. We hope this seems reasonable to you.
Comment 3: Provide references for ECOG-PS, Barthel and Karnofsky scores. The same for Apache-II and SOFA.
Response 3: The missing references for the different scores have been added.
Comment 4: Discuss (and reference) studies that are similar to this study.
Response 4: This issue has been improved and addressed in more detail in both the introduction and the discussion.
Comment 5: Artificial intelligence is having a huge impact on the ICU already, and could help solve several problems that are raised in this paper. This should be discussed in the discussion section.
Response 5: That is true. AI is currently changing daily clinical practice, so we considered appropiate to reflect your suggestion in our paper. Some paragraphs have been included in the discussion section with this intention.
Comment 6: Please correct the reference numbering.
Response 6: This mistake appeared when the pdf document was generated. It has been corrected.
Comment 7: Different font in lines 227-230.
Response 7: Thank you for noticing. It has been corrected.
Comment 8: Supplementary figure: either put it in the paper as figure 3, or put it in the supplementary material.
Response 8: We agree with your suggestion. The journal only accepted 2 figures, that is why we attached it as Supplementary material when we made the submission. If editors allow it, we could put it as figure 3.
Reviewer 4 Report
Comments and Suggestions for Authors
First, I would like to congratulate the authors on completion of this manuscript. it is extensively studied topic and not to surprise showing cancer patients especially with advanced disease do poorly if enough be in the ICU.
No major concern. Please add the limitation that most patients had stage 4 disease and more than half received chemo within 30 days. These two factirs will lead to poorer outcomes and skew the results
Comments on the Quality of English LanguageMinor revision
Author Response
Thank you very much for taking the time to review this manuscript. We are grateful for the favorable evaluation and the congratulations. As you suggested, we have improved the work limitations section by including your suggestions. Please find the detailed responses below and the corresponding revisions/corrections highlighted in the re-submitted files, as well as a clean version of the revised manuscript.
Point-by-point response to Comments:
Comment 1: Please add the limitation that most patients had stage 4 disease and more than half received chemo within 30 days. These two factors will lead to poorer outcomes and skew the results.
Response 1: Thank you for pointing this out. We agree with this comment. Therefore, we have improved the work limitations section by including your suggestions.
Round 2
Reviewer 1 Report
Comments and Suggestions for Authors
Thanks for your revised manuscript. You can see my comments below.
Authorship and institutions:
· The authors may need to use the Journal template to edit this part. Perhaps, the journal editing can fix it.
Introduction
· The in-text citation format is incorrect. It should be [e.g.,12-17] after the last word of sentence and after the closing brackets instead of using the superscripts.
· Add in text citation in “Mistakes regarding the decision if whether a patient would benefit or not from being admitted to an ICU can lead to losing options for cure or long survival in some patients with cancer, or, at the opposite end, applying futile treatments that cause unnecessary suffering at the end of life. Generally, ICU admissions are appropriate when the critical condition is potentially reversible with a reasonable probability, the oncological prognosis is long enough to justify aggressive therapies, other comorbidities do not advise otherwise, and the patient does not decline intensive care therapies after discussion with a trained physician if possible.”
Discussion
· “Actually, artificial intelligence (AI) has been successfully used in intensive care to develop continuous monitoring systems that can warn physicians about possible changes in a patient’s condition before they become critical. These systems can analyse data such as hear rate, blood pressure, blood oxygen levels and other parameters to identify early signs of deterioration and enable rapid intervention. This can be especially useful in emergency situations where rapid and accurate assessment is needed to make treatment decisions. In summary, AI in critical care has proven to be a valuable tool to improve the quality of care, reduce medical errors and save lives by enabling faster, more accurate and personalized care for critically ill patients. Since some of the risk factors for poor prognosis identified in our study are routinely reflected in medical records and would be easily interpretable by AI, this same technology could have applicability in this decision-making scenario, provided that the usefulness of these variables was corroborated in prospective cohorts of patients with cancer.”
This paragraph is not relevant to your study. It can be recommended but not suitable to introduce the AI that detail because it is not your study purpose. I suggest only a brief recommendation for future studies if AI is added in the treatment protocol.
Overall
English language editing is needed
Comments on the Quality of English LanguageMinor
Author Response
Summary
Thank you very much for taking the time to review the manuscript again. We are glad our improvement of the manuscript resolved most of the concerns. We here address the comments made in the second-round revision. We hope that our proposals resolve them satisfactorily and, therefore, improve the work. Please find the detailed responses below and the corresponding revisions/corrections highlighted changes in the re-submitted files, as well as a clean version of the revised manuscript.
Point-by-point response to Comments and Suggestions for Authors
Comment 1: Authorship and institutions: The authors may need to use the Journal template to edit this part. Perhaps, the journal editing can fix it.
Response 1: We would be very grateful if the journal editing team could fix it to avoid mistakes.
Comment 2: Introduction. The in-text citation format is incorrect. It should be [e.g.,12-17] after the last word of sentence and after the closing brackets instead of using the superscripts.
Response 2: That issue has been resolved.
Comment 3: Introduction. Add in text citation in “Mistakes regarding the decision if whether a patient would benefit or not from being admitted to an ICU can lead to losing options for cure or long survival in some patients with cancer, or, at the opposite end, applying futile treatments that cause unnecessary suffering at the end of life. Generally, ICU admissions are appropriate when the critical condition is potentially reversible with a reasonable probability, the oncological prognosis is long enough to justify aggressive therapies, other comorbidities do not advise otherwise, and the patient does not decline intensive care therapies after discussion with a trained physician if possible.”
Response 3: The text citation has been added.
Comment 4: Discussion. “Actually, artificial intelligence (AI) has been successfully used in intensive care to develop continuous monitoring systems that can warn physicians about possible changes in a patient’s condition before they become critical. These systems can analyse data such as hear rate, blood pressure, blood oxygen levels and other parameters to identify early signs of deterioration and enable rapid intervention. This can be especially useful in emergency situations where rapid and accurate assessment is needed to make treatment decisions. In summary, AI in critical care has proven to be a valuable tool to improve the quality of care, reduce medical errors and save lives by enabling faster, more accurate and personalized care for critically ill patients. Since some of the risk factors for poor prognosis identified in our study are routinely reflected in medical records and would be easily interpretable by AI, this same technology could have applicability in this decision-making scenario, provided that the usefulness of these variables was corroborated in prospective cohorts of patients with cancer.”
This paragraph is not relevant to your study. It can be recommended but not suitable to introduce the AI that detail because it is not your study purpose. I suggest only a brief recommendation for future studies if AI is added in the treatment protocol.
Response 4: Thank you for the suggestion. We have removed the paragraph from the manuscript, but will take into account the recommendation for future studies.
Comment 5: English language editing is needed.
Response 5: The first version of the manuscript was revised by a native translator. The new version has also been checked by a colleague fluent in English writing following your suggestion.
Reviewer 3 Report
Comments and Suggestions for Authors
All comments have been addressed.
Author Response
Thank you very much for your feedback.